# An Innovative Fiber-Optic Hydrophone for Seismology: Testing Detection Capacity for Very Low-Energy Earthquakes

**DOI:** 10.3390/s23073374

**Published:** 2023-03-23

**Authors:** Sergio Guardato, Rosario Riccio, Mohammed Janneh, Francesco Antonio Bruno, Marco Pisco, Andrea Cusano, Giovanni Iannaccone

**Affiliations:** 1Istituto Nazionale di Geofisica e Vulcanologia, Sezione di Napoli-Osservatorio Vesuviano, Via Diocleziano, 328, 80124 Napoli, Italy; 2Optoelectronic Division-Engineering Department, University of Sannio, c.so Garibaldi 107, 82100 Benevento, Italy

**Keywords:** bradyseism, earthquake recording, seismic monitoring, fiber-optic hydrophone, optical fiber technology

## Abstract

An innovative fiber-optic hydrophone (FOH) was developed and investigated via an experiment at sea; it is capable of operating at a very low frequency of the seismic spectrum and detecting small magnitude earthquakes. The FOH exploits an optical fiber coil wrapped around a sensitive mandrel in a Michelson interferometric configuration. The FOH operated for about seven days at a water depth of 40 m, in the Campi Flegrei volcanic area (Southern Italy), and a few meters from a well-calibrated PZT hydrophone used as a reference. Thirty-three local earthquakes occurred during the simultaneous operation of the two hydrophones, allowing a straightforward comparison of the recordings. The local earthquakes occurred at an epicentral distance less than 2.5 km from the site of recording, and were estimated to be in the range of magnitude from −0.8 to 2.7. The analysis of the recorded earthquake waveforms in the frequency and time domains allowed retrieving the response function of the FOH in the frequency range from 5 to 70 Hz. The FOH responsivity in terms of acoustic pressure reached about 230 nm/Pa and was flat in the studied frequency range. Due to the high quality of the FOH recordings, this equipment is suitable for applications addressing submarine volcanic activity and the background seismicity of active faults in the ocean.

## 1. Introduction

The background seismicity of active volcanoes is often composed of sequences of earthquakes known as seismic swarms, which occur in a small volume over a short time frame and with no typical mainshock. The Campi Flegrei volcanic area near Naples in Southern Italy is characterized by low-magnitude earthquakes and frequent seismic swarms. The area is monitored by a dense seismic network consisting of 33 stations, including broad-band seismometers, short-period sensors, and accelerometers. While this network typically provides accurate hypocenters and other seismic source parameters, the overlapping recordings during seismic swarms make interpretation difficult. Experiments using hydrophones have been conducted to record the local seismicity of the Campi Flegrei area, a partially submerged area forming the Gulf of Pozzuoli.

When recording an earthquake, the hydrophone produces a simpler and shorter trace than a seismometer, as it is only sensitive to longitudinal (P-wave) waves and does not register shear waves or most surface waves. Thus, during swarms, a hydrophone is more effective than a seismometer. In seismology, hydrophones with a frequency response extending below 100 Hz are required, as low-magnitude earthquakes producing background seismicity in active volcanic areas mainly emit high-frequency seismic body waves, unlike strong earthquakes, which produce seismic waves with periods of tens to hundreds of seconds. For example, seismic events with magnitude M < 4 produce elastic waves in the 1 to 70 Hz frequency band that are typically recorded at a distance of no more than a few hundred kilometers.

Piezoelectric ceramic hydrophones (PZTs) are widely used and considered the gold standard of underwater pressure sensing due to their excellent performance and ease of manufacture, but they require a power supply and preamplifier at the sensor location. In recent years, research has shifted towards developing new hydrophones based on optical fiber technology (FOH, fiber-optic hydrophone), which offer advantages such as small size, light weight, environmental ruggedness, and the ability to act as a transmission medium.

There are several proposals in the literature for FOHs with different transmission principles and performance. The first idea of using an optical fiber wound around a compliant mandrel to develop an acoustic hydrophone dates back to 1977 [1]. In the following years, many research groups have proposed various configurations of fiber-optic sensors, such as interferometric systems [2], coated FBG-based sensors [3,4], and fiber lasers [5], to develop hydrophones for underwater applications. In example, more authors demonstrated an interferometric hydrophone based on a Fabry–Perot reflector consisting of a photonic crystal reflector suspended on a single-mode fiber tip [6]. Other authors [7] developed a distributed fiber laser feedback hydrophone (DFB-FL) and deployed it on the seafloor for a field test on the south coast of Australia [8]. In 2014, a linear towed array with four fiber laser hydrophones using a DFB-FL was used as the sensing element [9] and was tested in Mogan Mountain lake. More recently, various configurations of interferometric FOH have been proposed [10,11,12]. Among these, a fiber-optic towed array operating in an acoustic frequency range up to 10 kHz was recently demonstrated [13]. Lavrov et al. conducted experimental trials of FOHs involving either FBGs or Faraday rotating mirrors in a lake in Russia [14,15].

The performance of FOHs is generally comparable to that of piezoelectric hydrophones for a wide range of applications, including underwater target detection [16] and oil and natural gas prospecting [17]. For seismological applications, the capacity to record signals of frequencies lower than 100 Hz is required. High-sensitivity FOH hydrophones have been proposed in this frequency band, but with underwater tests developed in the laboratory without applications at sea [18]. For these reasons, a new hydrophone using interferometry for wave modulation has been developed in the framework of the OPTIMA project, funded by local authorities of the Regione Campania in Southern Italy [19]. This hydrophone is capable of measuring weak hydro-acoustic pressure perturbations with high sensitivity in a frequency range of interest in seismology (<100 Hz).

The OPTIMA project (2018–2022) aimed to carry out research activities to demonstrate the potential of optoelectronic technology for the creation of new sensor systems for the detection of parameters of interest in the marine environment and for medical applications. In particular: (1) measurement of physical and biological quantities for monitoring the state of sea water, studying marine fauna, geophysical and volcanic monitoring, and monitoring of maritime traffic; and (2) demonstration that the use of biosensors can result in industrial, commercial, and scientific advantages in terms of miniaturization of the devices, reduction in the number of reagents, enhanced accuracy and repeatability of the measurements, and rapid availability of the results.

This paper describes the design and development of the new FOH, and the seven-day field test performed in the marine sector of the Campi Flegrei volcanic area. The analysis of 33 local earthquakes, recorded simultaneously by the new FOH hydrophone and a well-calibrated PZT hydrophone used as a reference, are presented. Finally, the results of the analysis in terms of frequency band and responsivity, and potential fields of application, are discussed.

## 2. Materials and Methods

### 2.1. The Fiber-Optic Hydrophone

The developed fiber-optic hydrophone is one in which two layers of optical fiber are wound around a composite cylinder to create a transfer of strain. The schematic representation of the sectional and lateral view of the FOH are shown in Figure 1.

The composite cylinder consists of compliant layers (mandrel, oil-filled, and cork) and a supporting steel rod; the optical fiber runs all around the cylinder. When the FOH is subjected to an acoustic pressure (P) with a wavelength much longer than the FOH size, such as that of a converted wave generated by a local earthquake, a uniform force is exerted on the external boundaries of the FOH cylindrical structure, leading to a radial deformation of the complaint mandrel layer. The expansion (or compression) of the cylindrical mandrel induces a strain in the optical fiber wrapped around it. This fiber deformation can be determined using a Michelson interferometric scheme, which comprises a sensing arm of the optical fiber wrapped around the FOH of length $l_{f}$ and a reference arm “dummy” FOH insensitive to pressure. Two Fiber Bragg Gratings (FBGs), operating at different wavelengths, are used to mark the beginning and the end of the fiber wrapped around the mandrel. In conjunction with an interferometric technique, the FBGs, acting as reflectors, can be used to detect the elongation of the fiber coil. A commercial interrogator system (MultiZonaSens, Optics11) then provides an output signal directly proportional to the changes in the fractional length versus time.

The responsivity $R_{l}$ of the FOH under the applied pressure can be expressed in terms of fractional change in fiber length $\Delta l_{f}$ with respect to the applied pressure P as follows:$$R_{l}=\frac{∆l_{f}}{P}$$

The responsivity evidently depends on the combination of the elastic and geometric properties of the FOH. The novel FOH was designed [18] to operate at low frequencies (below 100 Hz) and was fabricated in a rugged fashion, equipped with pressure hydraulic compensation mechanics for underwater operation. Because of consideration of environmental noise, and the noise of the interrogation system itself (intrinsic to the system composed of the optical fiber and the electronics that convert the light to an electric voltage signal followed by an ADC), a target responsivity of 346 · 10^6^ nm/µPa was estimated (see Section 3).

The final configuration of the fiber-optic system consisted of a FOH used as the main (sensitive) gauge and a dummy hydrophone acting as an optical reference (DOH). Figure 2 shows the fabricated sensing (brown color) and reference dummy (silver color) FOHs. The optical reference hydrophone consists of a solid mandrel on which the optical fiber is wound. A hollow cylindrical metal shell surrounds the solid mandrel, providing further protection of the optical fiber from mechanical damage and incoming acoustic pressure waves. The rigid supports on both sides also provide fiber passages. Two nuts are used on both sides of the rod to lock the reference hydrophone. A clamped region was also created in this case. The size of the reference hydrophone is smaller than that of the sensitive hydrophone. Indeed, the reference hydrophone has a diameter of 8.9 · 10^−2^ m and a height of 12.1 · 10^−2^ m, while the main hydrophone height is 25.4 · 10^−2^ m. Essentially, the overall structure does not allow transfer of strain resulting from an incoming pressure wave to the reference optical fiber, because there are no sensitive parts. The reference hydrophone was deployed in the same operative conditions as those of the optical hydrophone.

For redundancy and multiplexing proof, two pairs composed of the sensitive and dummy hydrophones were then installed and integrated on the seafloor multi-parametric module used in our final system architecture, and fixed to the seabed.

### 2.2. The PZT Reference Hydrophone

The hydrophone used as a reference (RH) is a passive ultra-low frequency sensor hydrophone with a response function between 0.01 Hz and 8 kHz, i.e., the HTI-04-PCA/ULF model of High Tech Inc. (Long Beach, MS, USA). (http://www.hightechincusa.com/products/hydrophones/hti04pcaulf.html (accessed on 21 March 2023)), coupled with an external transconductance pre-amplifier with a gain of 26 dB. The HTI-04 features an oil-filled pressure-compensated design capable of operating at full ocean depth (6000 m), and is used in the anti-submarine warfare (ASW) industry and for noise measurements with a receiving sensitivity of −194 dBV regarding a sound pressure level (SPL) of 1 µPa (@ T = 20 °C) at a 1 m distance.

The hydrophone is provided with a bulkhead cable on the body (butyl rubber boot encapsulated material) and a wet-mate connector, having a total size of a 17.3 · 10^−2^ m length and a 5.1 · 10^−2^ m diameter.

The HTI-04 hydrophone is based on the piezoelectric effect (PZT) acting over the piezoelectric ceramic cylinders, which form the hydrophone active acoustic element. The ceramic cylinders are radially polarized (omnidirectional sensitivity for frequency <8000 Hz), and, with end caps, the enclosed cylinder forms a small pressure vessel. The HTI-04 is a passive sensor with a single-ended output signal connected to at an external preamplifier, designed around the AD795J operational amplifier. This is a low-power and low-noise precision FET op-amp used in the non-inverting second-order Sallen–Key low-pass active filter configuration with a −3 dB point at about 120 Hz. The preamplifier itself is a transconductance amplifier design, which eliminates the need for separate power and signal conductors. Preamplifier power consumption is less than 33 mW using a 12 V dc dual-output power supply. Finally, the output signal is acquired by a Quanterra-Kinemetrics Q330 data logger at a sample rate of 200 sps, with no input gain.

The sensitivity of the RH was provided by the manufacturer as a unique value valid within the sensor’s frequency range from 0.01 Hz to 8 kHz. It is worth remarking that the calibration of a hydrophone below 100 Hz is known to be a difficult task, which requires complex procedures and facilities that are not always available [19,20]. An innovative in situ comparative procedure, using a calibrated broad-band seismometer (Trillium Compact-OBS, Nanometrics), was developed to verify the validity of the calibration value of the reference hydrophone at low frequencies. The equivalence of the waveforms of the earthquake-induced ground acceleration and the water pressure, in the case of the co-location of a hydrophone and a seafloor seismometer in shallow water, was demonstrated in a previous paper [21]. The study used recordings of a set of local earthquakes with epicentral distances of a few tens of kilometers, and regional earthquakes with a wide range of magnitude (2.7 < M < 6.8) recorded by a seismometer and the RH installed at the same site and used for the present paper. In particular, the derivative of the vertical component of the seismometer recordings was evaluated in order to obtain the acceleration signal used as a reference with the RH recordings, in terms of pressure variation according to the sensitivity value provided by the manufacturer. The similarity of amplitudes and frequency content at low frequencies (down to 0.1 Hz) of the signals recorded by the two instruments for all the analyzed earthquake records, as measured by a high value of the correlation coefficient, demonstrates the validity of the calibration value used for the RH.

### 2.3. Sea Trial and Experimental Setup

The tests of the FOH operation and performance were developed at the marine MEDUSA infrastructure, which is part of the geophysical monitoring system of the Campi Flegrei volcanic area [22]. MEDUSA (Multiparametric Elastic-Beacon-Based Devices and Underwater Sensor Acquisition) is a permanent network of four marine multidisciplinary platforms (MMPs) operating since 2016 in the Gulf of Pozzuoli, which supplement the land-based geophysical monitoring network managed by the Istituto Nazionale di Geofisica e Vulcanologia (INGV) (Figure 3a). Each MMP consists of an anchored spark buoy connected by cable to an instrumented module on the seabed, placed a few meters from the ballast of the buoy (Figure 3b).

The seafloor module is equipped with geophysical and oceanographic sensors: (i) a 0.01 ÷ 50 Hz seismometer; (ii) a hydrophone, namely the reference hydrophone RH; (iii) a Paroscientific high-resolution pressure sensor; and (iv) a single-point three-component acoustic current meter. A cylindrical subsea pressure vessel contains the acquisition systems and the electronic devices. The acquisition of the four data streams produced by the three components of the seismometer and by the hydrophone is performed with a sampling rate of 200 sps for the RH by means of a Quanterra-Kinemetrics Q330 digitizer.

The emerging part of the buoy hosts 5 GHz and UMTS-4G communication systems, solar panels, a weather station, and control electronics. In addition, on the top of the buoy a geodetic GNSS receiver monitors the seafloor movements produced by the volcanic activity of the area [23,24].

The electro-mechanical cable connecting the seafloor module to the buoy supplies power, data transmission, and a clock signal from the GPS receiver.

A dedicated submarine frame was developed to host the sensor for the deployment, near one of the MEDUSA seabed modules, for the specific purpose of the FOH test. The FOH frame is smaller than the MEDUSA seabed modules and is made of stainless steel with dimensions of 0.51 m × 0.81 m. To synchronize the recorded signals, RH and FOH used the same clock signal, locked to the GPS. To maximize the probability of recording a significant number of earthquakes, the FOH module was deployed in the proximity of the MMP (CFB3), the closest MEDUSA platform to the area where earthquake occurrence is the most frequent, that is, the Solfatara crater [25].

Figure 4 shows the FOH module (b) deployed and positioned on the seafloor by means of an electromechanical cable, at a distance of about 2 m from the existing MMP at the CFB3 site (a).

The optical interrogator of the FOH was mounted on the top of the corresponding buoy (Figure 4a) and connected to the FOH by means of a 70 m long optical fiber cable for the data transmission.

The frame hosting the FOH was installed on 14 December 2020, and operated for 7 days.

In the following, all the data analysis presented was performed using earthquake recordings from the FOH and RH.

## 3. Results and Discussion

The local seismicity bulletin produced by INGV-OV reports the events occurring during the test period of the FOH. From this list, events according to the quality of their recordings in terms of signal-to-noise ratio were selected. In particular, the primary selection criterion applied was the adoption of a threshold value of this ratio set to 5, measured over a time window of 2 s starting from the P arrival time. Thirty-three events were finally selected according to this criterion. The corresponding hypocentral parameters are reported in Table 1 and Figure 5.

The earthquakes had a magnitude range of −0.8 < Md < 2.7 and mostly occurred around the Solfatara crater (Figure 5), as expected. Their epicentral distance from the RH and FOH located at the CFB3 site was less than 2.5 km and the hypocentral depths were within the first 3 km. Unfortunately, no stronger earthquakes occurred at regional distances (greater than 100 km).

For the sake of brevity, this section examines the results of the analysis applied to one earthquake that can be considered an appropriate example to analyze the signal in the frequency domain. The complete analysis of the 33 earthquake recordings is shown in the Appendix A.

Figure 6 shows the waveform and the amplitude spectra of the earthquake recorded by both the FOH and the RH. The signal of the RH was converted into physical units of pressure using a conversion factor of 6.28 · 10^−4^ Pa/count to take into account the data acquisition system and the sensitivity. The signal of the FOH is reported in counts.

From Figure 6, a long-period waveform is evident; this is due to the overlapping of the pressure variation induced by the sea waves and the earthquake signal, and corresponds to a large spectral amplitude value in the 0.5 ÷ 5 Hz frequency band. On the contrary, this characteristic is not present in the lower panels of the FOH. At frequencies greater than 5 Hz, the two amplitude spectra show very similar behavior and this feature can be observed on the spectra of the other analyzed events reported in the Appendix A. This is evidence that the frequency response of the FOH is limited at a low frequency of about 5 Hz.

The transfer function required to convert the amplitude count values recorded by the FOH in pressure units (Pascal) was calculated from the frequency domain amplitude ratio between the signals of the same earthquake recorded by the FOH and RH. A smoothing-filtering method was applied to both the spectra, to average out the spectral fluctuations, according to the processing procedure described in [26].

Figure 7 shows the transfer function, or acoustic pressure responsivity, obtained from the spectra ratios of the 33 selected earthquake records reported in the Appendix A.

The blue curve represents the median of all 33 spectral ratios used. It shows a flat trend with a strong low-frequency decay where 5 Hz represents the corner frequency value. As shown in Figure 7, it is remarkable that all curves are included in a narrow band of ±50 nm/Pa around the median value.

The mean responsivity value in the 5 ÷ 40 Hz frequency band was used in order to compare the full waveform of the FOH recordings with the RH; in this frequency band most of the energy of the recorded earthquakes is present, as shown from the amplitude spectra. Then, the value of 230 nm/Pa was considered to convert the amplitude values of the FOH recorded signals into pressure values of Pa. A final check of this value was obtained by computing the cross-correlation coefficient (R) between the earthquake records of the two hydrophones. The record pairs were previously high-pass filtered at 5 Hz with a Butterworth filter. As an example, Figure 8 shows an interesting record in which two consecutive earthquakes occurred within a short time. Here the records of the two hydrophones are displayed superimposed on the same plot. A clear similarity of the two waveforms is evident through the complete and persistent eight-second overlap of the waveform duration.

The highest value of the cross-correlation coefficient (0.93) shown in Figure 8 confirms that the two waveforms have the same frequency content, both in amplitude and phase. The same analysis performed in the frequency band of 5 ÷ 70 Hz on the records of the 33 selected earthquakes confirms the highest value of the R coefficient for all the analyzed pairs (Table 1 and Figure 9).

Figure 9 and Table 1 show that the lowest value of R is 0.69 and most of the events have a R value greater than 0.75 (31 events out of 33). There is no evident relationship between R and Md, i.e., high values of R correspond to a wide range of magnitudes. This indicates excellent recording quality with the FOH, even for events of very small magnitude, including less than zero.

## 4. Conclusions

A new fiber-optic hydrophone was successfully developed and investigated via an experiment at sea, taking advantage of MEDUSA marine facilities in the Gulf of Pozzuoli. The FOH was installed on a new dedicated frame and deployed at a depth of 40 m, close to a reference well-calibrated hydrophone RH on board an existing seafloor instrumented module. During the sea trial period (seven days), 33 earthquakes with magnitudes ranging from −0.8 to 2.7 occurred in a small volume of the Solfatara area, at a distance not greater than 4 km from the FOH and RH. As the seismic waves’ path from the source to the recording site was similar for all earthquakes, the seismic wave attenuation was disregarded and the magnitude considered to be the sole parameter reflecting the energy and amplitude of the radiated seismic waves. According to the magnitude range of the selected earthquakes, the frequency content of the seismic waves was expected to range from a few hertz to 40 ÷ 50 Hz [27,28]. Nevertheless, below this frequency range, wind-generated acoustic waves and microseisms could also be observed. The top panels of Figure 6 show an example of the low-frequency content of the analyzed acoustic waves. In fact, the waveforms recorded by the RH show the presence of a long-period signal and an evident and sharp peak in the amplitude spectrum. On the contrary, the low-frequency waves and the corresponding spectral peak are completely absent in the FOH recordings of Figure 6 (bottom panels). The FOH/RH spectral ratio derived for all the recorded earthquakes shows that the low-frequency response cut-off of the FOH is 5 Hz (Figure 7). For frequencies greater than 5 Hz, Figure 7 shows an almost flat trend of the FOH/RH ratio, with a mean value of 230 nm/Pa. A perfect match between the recordings of FOH and RH was achieved using this value as the conversion factor of the seismic recordings of the FOH into pressure units (Figure 8 and Appendix A).

The FOH trial shows that in the frequency and amplitude range of interest of seismic activity monitoring, the FOH has an excellent frequency response, equivalent to that of one of the most common high-quality sensors, namely, the HTI-04-PCA/ULF. The results of the comparison between the FOH and RH indicate the excellent suitability of the FOH in the standard seismic frequency range with high sensitivity and high fidelity.

## Figures and Tables

**Figure 1 sensors-23-03374-f001:**
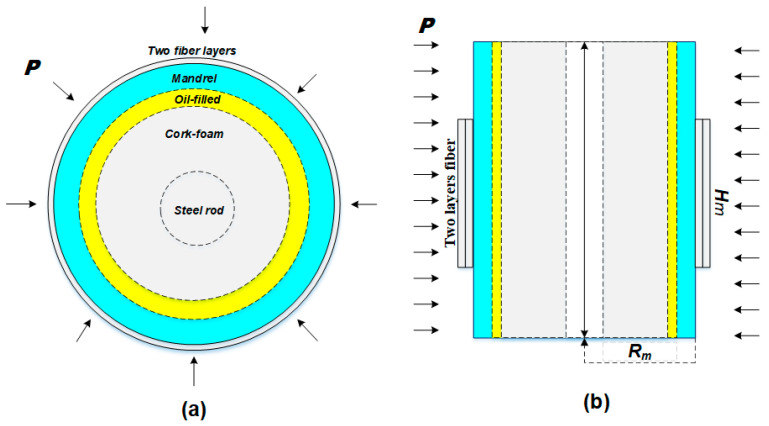
(**a**) Cross-section view of the composite FOH and (**b**) lateral section of the FOH. $R_{m}$ and $H_{m}$ represent the mandrel radius and height of the two overlays formed by the wrapped optical fiber, respectively, while the rectangle on both sides and P (with arrows) represent the transverse section of optical fibers forming a double layer of fibers wound around the mandrel and the underwater pressure on the external boundaries of the FOH.

**Figure 2 sensors-23-03374-f002:**
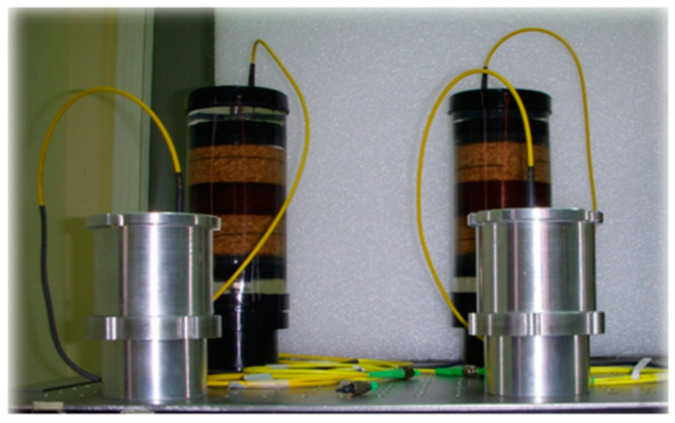
A photo of the fabricated sensing (brown color) and reference dummy DOH (silver color) FOHs.

**Figure 3 sensors-23-03374-f003:**
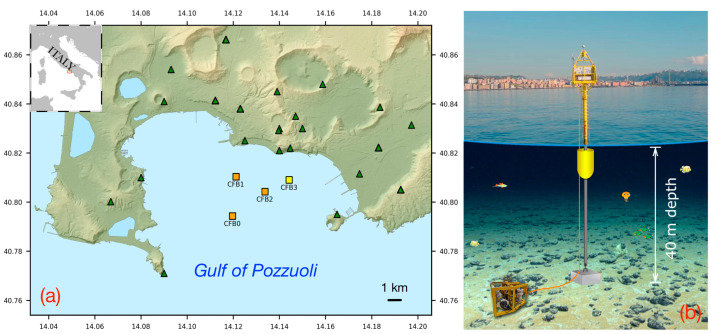
(**a**) Map of the seismic monitoring network of the Campi Flegrei volcanic area. The locations of the buoys of the MEDUSA infrastructure are represented by the orange and yellow triangles. Green triangles represent the on-land seismic stations. (**b**) Sketch of the marine multidisciplinary platform. (The reported sea depth refers to the CFB3 module only).

**Figure 4 sensors-23-03374-f004:**
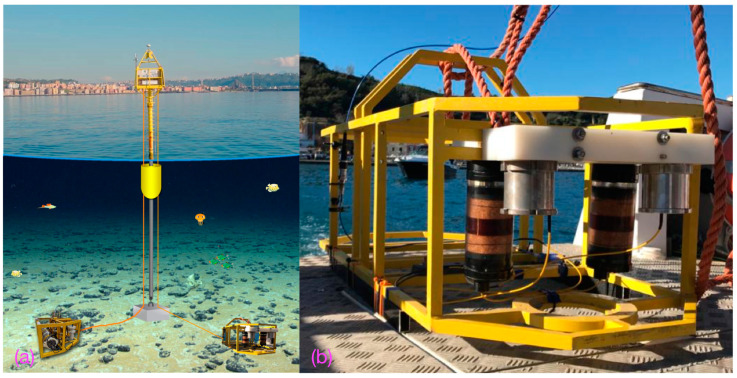
Sketch of the marine multidisciplinary platform used to test the FOH. In (**b**) the new seafloor module hosting the FOH (and at the bottom right of (**a**)), and the instrumented module CFB3 installed at the seafloor (**a**). (Drawing is not to scale).

**Figure 5 sensors-23-03374-f005:**
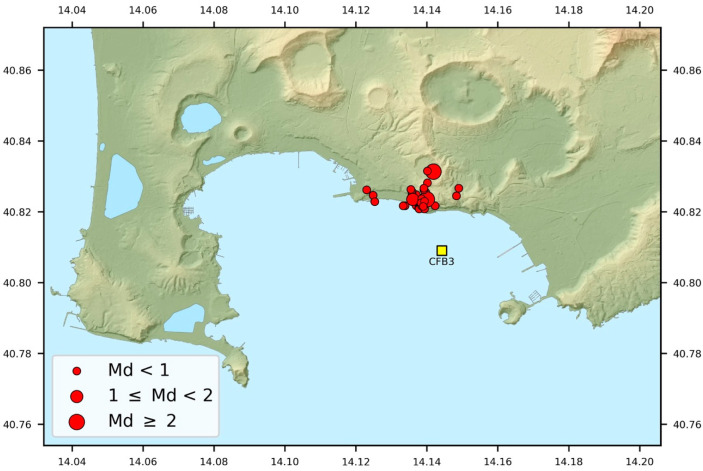
Map of epicenters of earthquakes used in this study. Dimensions of the circles are proportional to magnitude. The yellow square represents the location of the CFB3 instrumented module at the seabed of the MEDUSA infrastructure.

**Figure 6 sensors-23-03374-f006:**
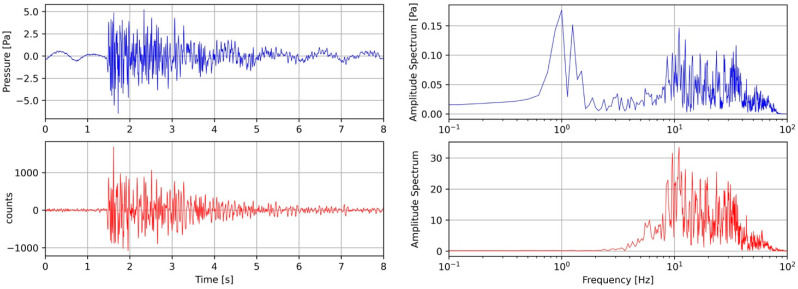
Waveforms (R = 0.7) and amplitude spectrum of the Md = 0.8 earthquake occurring on 2020-12-20 03:17:45 UTC recorded by the reference RH (blue color) and FOH (red color) hydrophones.

**Figure 7 sensors-23-03374-f007:**
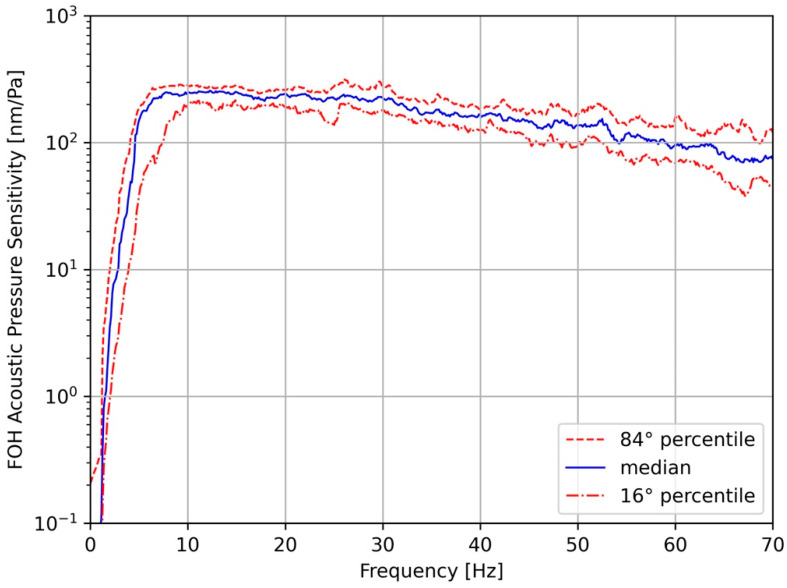
FOH acoustic pressure sensitivity curves obtained by the amplitude spectra ratios of the 33 analyzed earthquakes. The solid blue line is the median of the distribution. The red dashed lines bound the 16th–84th percentile range.

**Figure 8 sensors-23-03374-f008:**
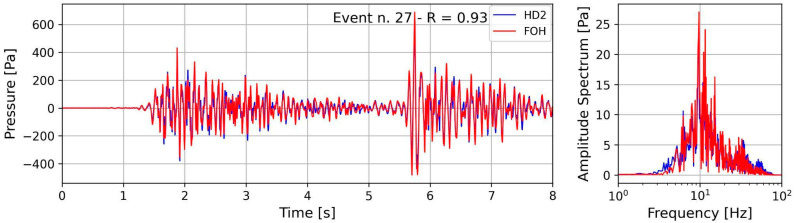
Waveforms and amplitude spectra of the earthquake (Event) n. 27 (see Table 1) recorded by the two hydrophones, RF blue color and FOH red color. Both records are high-pass filtered at 5 Hz. The report value R = 0.93 represents the correlation coefficient between the two traces.

**Figure 9 sensors-23-03374-f009:**
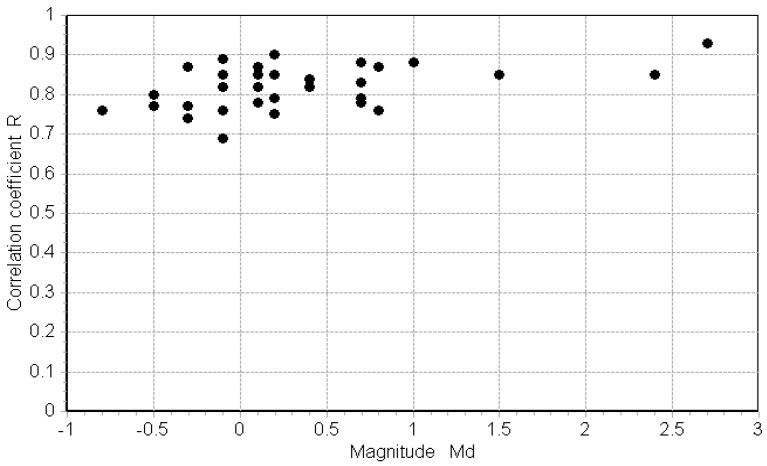
Plot of the correlation coefficient R vs. the magnitude Md value.

**Table 1 sensors-23-03374-t001:** List of the earthquakes used for the analysis. The R column reports the correlation coefficient between the records of the earthquake provided by the reference and fiber-optic hydrophones.

Event N#	Origin Time (UTC)	Md	R
1	2020-12-14 21:10:17	0.2	0.79
2	2020-12-16 14:28:20	0.1	0.78
3	2020-12-17 03:00:09	−0.3	0.87
4	2020-12-17 03:11:29	−0.1	0.89
5	2020-12-17 22:18:58	−0.3	0.74
6	2020-12-19 00:43:49	−0.5	0.77
7	2020-12-19 06:29:28	0.2	0.90
8	2020-12-19 06:30:04	0.1	0.85
9	2020-12-19 14:09:57	−0.1	0.82
10	2020-12-19 14:16:24	0.2	0.75
11	2020-12-19 14:17:00	0.7	0.88
12	2020-12-19 15:11:20	−0.1	0.82
13	2020-12-19 15:12:49	0.1	0.87
14	2020-12-19 15:17:56	−0.3	0.77
15	2020-12-19 15:18:11	0.4	0.84
16	2020-12-19 15:20:25	0.2	0.85
17	2020-12-19 15:21:42	1.0	0.88
18	2020-12-19 15:32:51	−0.1	0.69
19	2020-12-19 15:47:02	0.8	0.87
20	2020-12-19 16:44:27	0.1	0.78
21	2020-12-19 17:08:32	0.7	0.83
22	2020-12-19 17:09:05	−0.1	0.85
23	2020-12-19 17:11:21	−0.5	0.80
24	2020-12-19 20:47:54	−0.1	0.76
25	2020-12-19 20:48:59	0.7	0.78
26	2020-12-19 21:16:08	0.4	0.82
27	2020-12-19 21:54:53	2.7	0.93
28	2020-12-20 00:30:27	−0.8	0.76
29	2020-12-20 02:53:48	0.7	0.79
30	2020-12-20 03:13:49	2.4	0.85
31	2020-12-20 03:15:11	0.1	0.82
32	2020-12-20 03:17:45	0.8	0.76
33	2020-12-20 03:20:27	1.5	0.85

## Data Availability

Data available on request.

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
