# Peer review of "An Innovative Fiber-Optic Hydrophone for Seismology: Testing Detection Capacity for Very Low-Energy Earthquakes"

_sensors, 2023, doi:10.3390/s23073374_

Round 1

Reviewer 1 Report

In this study, an experimental investigation is presented to detect small magnitüde eartquakes through fiber optic hydrophone. The presented approach seems innovative and the text is overall well written. It can be accpeted for publication after minör revisiions. My comments can be found below:

·     -    Please use Passive Voice in the text instead of using “We”.

·      -    More extensive information can be provided regarding the OPTIMA Project.

·       -  At the last paragraph of Introduction section, please remove the expression of “Please highlight 68 controversial and diverging hypotheses when necessary.”

     - Please add equation number in the text.

·    - More comments should be provided in the section 2.3 regarding the experimental setup.

Author Response

Si prega di consultare l'allegato. Si prega di consultare l'allegato.

Reviewer 2 Report

The proposed work, "An innovative fiber optic hydrophone for seismology: 2 testing detection capacity for very low-energy earthquakes," is reported with practical results and deployed on-site.

It is recommended for acceptance for publication with the following minor revisions.

It is advised authors enhance the introduction section by adding more recently deployed works reported on fiber optic hydrophones for seismology.

Advised authors to add a performance comparison table of parameters between proposed and more similar recent works reported

Check for  grammatical errors in the whole manuscript. 

Reviewer 3 Report

Article “An innovative fiber optic hydrophone for seismology: testing detection capacity for very low-energy earthquakes” discussed using fiber optic hydrophone as seismic meter, which is capable of operating at very low frequency of the seismic spectrum and for detecting small magnitude earthquakes. The recorded earthquake waveforms in the frequency and time domain showed the response function of the FOH in the frequency range from 5 to 70 Hz, which means that the equipment is suitable for applications addressing submarine volcanic activity and the background seismicity of active faults in the ocean.

I think this article is suitable for publication on "sensor"。
